# *Fusarium* spp. Associated with *Dendrobium officinale* Dieback Disease in China

**DOI:** 10.3390/jof8090919

**Published:** 2022-08-29

**Authors:** Seyed Ali Mirghasempour, Themis Michailides, Weiliang Chen, Bizeng Mao

**Affiliations:** 1Institute of Biotechnology, Zhejiang University, Hangzhou 310058, China; 2Kearney Agricultural Research and Extension Center, University of California Davis, 9240 South Riverbend Ave, Parlier, CA 93648, USA

**Keywords:** dieback, MLSA, morphology, tiepishihu

## Abstract

A rare plant species of the *Orchidaceae* family, *Dendrobium officinale* is considered among the top ten Chinese medicinal herbs for its polysaccharide. Since 2021, when the dieback disease of *D. officinale* was first reported in Yueqing City, Zhejiang Province, China, *Fusarium* isolates (number = 152) were obtained from 70 plants in commercial greenhouses. The disease incidence ranged from 40% to 60% in the surveyed areas. Multilocus sequence analysis (MLSA) coupled with morphological characterization revealed that the collected isolates belonged to five species (sp.), viz., *Fusarium concentricum*, *F. fujikuroi*, *F. nirenbergiae*, *F. curvatum*, and *F. stilboides*, with isolation frequencies of 34.6%, 22.3%, 18.4%, 13.8%, and 10.5%, respectively. Notably, at least two *Fusarium* species were simultaneously isolated and identified from the infected plants. Finally, the pathogenicity test results demonstrated that such species were responsible for the dieback disease of *D. officinale*. However, *F. concentricum* and *F. fujikuroi* were more invasive compared to the other species in this study. To the best of the authors’ knowledge, this study was the first report of *F. concentricum*, *F. curvatum*, *F. fujikuroi*, *F. nirenbergiae*, and *F. stilboides* causing the dieback disease of *D. officinale* in China and worldwide. This work provides valuable data about the diversity and pathogenicity of *Fusarium* populations, which will help in formulating effective strategies and policies for better control of the dieback disease.

## 1. Introduction

Commonly known as Tiepishihu, *Dendrobium officinale* (Kimura and Migo) is an epiphytic, herbaceous, flowering, insect-pollinated, and perennial plant with a cylindrical-fibrous stem that is mostly utilized for pharmaceutical purposes in traditional Chinese medicine (TCM) [1,2,3]. Thus far, it has been demonstrated that several bioactive constituents, comprising polysaccharides, alkaloids, phenanthrenes, bibenzyls, saccharides, glycosides, lignans, phenolic acids, and phenylpropanoids, possess a wide variety of pharmacological properties, including antioxidant, antitumor, and hypolipidemic activities, as well as anti-fatigue, hypoglycemic, anti-fibrotic, hepatoprotective effect, and immuno-enhancement enhancement effects, along with outcomes for rheumatoid arthritis and diabetes [1,3,4].

Tiepishihu is extensively cropped in several provinces of China including Zhejiang, Anhui, Fujian, Guizhou, Guangxi, Sichuan, and Yunnan, with a total cultivation area of nearly 4000 hectares [3,4,5], and it is valued between $450 and $3100 per kg. Recently, micropropagation and greenhouse farming technologies have been utilized to enhance the rate of low natural regeneration of this plant species [2,3,5]. The seedlings derived from tissue cultures are thus transplanted from March to May in greenhouses to produce tillers, and the flowers emerge about 15 months later, between May and July. The stems are further harvested from the fields from the end of October to the beginning of March, approximately 31 months after planting. For three years after the first harvest, the plant can also produce commercial yields (i.e., stems) annually.

Currently, *D. officinale* production is seriously threatened by a number of fungal genera such as *Alternaria alternata*, *Ceratobasidium* sp. (species), *Cladosporium cladosporioides*, *Colletotorichum gloeosporioides*, *C. fructicola*, *Epicoccum sorghinum*, *Fusarium equiseti*, *F. kyushuense*, *F. oxysporum*, *Neopestalotiopsis clavispora*, *Phoma multirostrata*, and *Sclerotium delphinii* [6,7,8,9,10,11,12]. Among them, *F. equiseti* causes the dieback disease, which was initially identified in Fujian Province, China [8]. The pathogen results in considerable losses of approximately 50%. The infected plants exhibit wilting and dieback on apical meristem leaves, followed by drying and death of the tips [8].

Phytopathologists and clinical microbiologists have now devoted much attention to *Fusarium* (Wollenweber and Reinkin; *Gibberella* as sexual morphs) as a species-rich, heterogeneous, and ubiquitous genus of filamentous fungi [13,14,15]. In addition, the fusarioid taxa are able to adapt to a variety of climatic zones and colonize a wide range of ecosystems and hosts [13,16]. To date, this agriculturally and clinically important genus is estimated to be composed of over 400 phylogenetically distinct species, 23 monophyletic species complexes, and quite a few monotypic lineages [13,15,17]. The *F. buharicum* species complex (FBSC), *F. fujikuroi* species complex (FFSC), *F. incarnatum-equiseti* species complex (FIESC), *F. lateritium* species complex (FLSC), *F. oxysporum* species complex (FOSC), *F. redolens* species complex (FRSC), *F. sambucinum* species complex (FSaSC), and *F. solani* species complex (FSSC) have been accordingly reported to cause devastating plant diseases over the years [13,17,18,19]. Moreover, it has been difficult to discriminate closely related *Fusarium* species through macro-/microscopic characteristics due to the high morphological variability inter-/intra-species. Nowadays, several new species have been further delineated within the *Fusarium* spp. complexes by applying a polyphasic approach to taxonomy that combines morphology-based identification and genealogical concordance among portions of multiple phylogenetically informative genes (i.e., genealogical concordance phylogenetic species recognition: GCPSR *sensu*). This technique has thus led to a major improvement in the Fusarioid fungi taxonomy and nomenclature [13,16,17,18,19,20].

The dieback disease is spreading in different Tiepishihu plantation areas, but there are no effective and eco-friendly control measures available. In this context, integrated disease management strategies, including biological control agents, biofertilizers, chemical fungicides, cultural practices, resistant varieties, forecasting models, and sanitation methods are required to minimize the incidence rate of plant pathogens and maintain profitable, sustainable Tiepishihu production [21,22].

Hitherto, little has been known about the species, or about the population structure and phenotypic characters of *Fusarium* species causing the dieback disease of *D. officinale*, neither in China nor elsewhere in the world. Against this background, this study aimed to identify and characterize the *Fusarium* spp. diversity, and then to assess the virulence of the disease for Tiepishihu.

## 2. Materials and Methods

### 2.1. Sample Collection, Fungal Isolation, and Morphological Characterization

The *D. officinale* plants (70) with dieback symptoms were initially sampled from commercial greenhouses in Yueqing (28.07° N, 120.57° E), Zhejiang Province, China. The incidence rate of the disease was assessed by visual observations, and then calculated for the presence or absence of symptomatic plants in the surveyed greenhouses. Afterward, the symptomatic stem tips were cut with a sterilized scalpel, superficially disinfected with a 2% solution of sodium hypochlorite (0.1% active ingredient of chlorine; [23]) for 1 min and 75% ethanol for 30 s, rinsed thrice with sterile distilled water, air dried on sterile filter papers under aseptic conditions, and finally placed onto potato dextrose agar (PDA) medium. The plates were subsequently incubated at 25 °C in the dark, and the colonies were purified by the hyphal tip method and then sub-cultured on the PDA and carnation leaf agar (CLA) media for morphological observation [13,24]. The conidial morphology and sporulation of the pure fungal colonies were finally examined under a Nikon Eclipse microscope (Japan).

### 2.2. DNA Sequencing and Molecular Phylogeny

DNA was extracted from the mycelia of 7-day-old cultures of the representative isolates using the Plant Genomic DNA kit (Tiangen, China) according to the manufacturer’s instructions. The fragments of the translation elongation factor 1-alpha (*tef1*), second largest subunit of RNA polymerase II gene (*rpb2*), and β-tubulin (*tub2*) genes were then amplified by the primers EF-1/EF-2, RPB2-5f2/RPB2-7cr, and Tub2F/Tub2R, respectively [13,23]. The polymerase chain reaction (PCR) was also performed in 25 μL volumes, containing 1 µL of genomic DNA, 12.5 μL 2 × Phanta^®^ Flash Master Mix Dye Plus (Vazyme, Nanjing, China), 9.5 μL of DNase-free water, and 1 μL of each forward and reverse primer (10 μM). Notably, the cycling conditions included the initial denaturation of 30 s at 98 °C, followed by 30 cycles of denaturation at 98 °C for 10 s, the annealing at 52 (*tef1*), 59 (*rpb2*), and 55 (*tub2*) for 10 s, the extension of 10 s at 72 °C, as well as the final extension at 72 °C for 1 min. The PCR products were first visualized on a 1% (*w/v*) agarose gel, and then Sanger sequencing was conducted by Sangon Biotech Co., Ltd. (Shanghai, China) for both directions to ensure high accuracy. The accession number of all generated sequences in this study was further obtained from the GenBank, as listed in Table 1. The aligned sequences of the novel isolates were also subjected to the Basic Local Alignment Search Tool (BLAST) to collect related sequences for inclusion in phylogenetic analysis. The BLASTN searches were fulfilled using the *rpb2*, *tef1*, and *tub2* sequences against the Nucleotide collection (nr/nt) database by restricting the material type. Multiple sequence alignments were correspondingly inferred in Molecular Evolutionary Genetics Analysis (MEGA) X software version 10.2.4 [25] using the MUSCLE (multiple sequence comparison by log-expectation) program [26] and refined manually if necessary. To generate concatenated datasets, single-gene sequences (*tef1*, *rpb2*, and *tub2*) were manually combined utilizing the BioEdit version 7.1 [27]. The phylogenetic trees were further constructed based on the individual and concatenated sequences (*rpb2*, *tef1*, and *tub2*) using the MEGA X software. The maximum likelihood (ML) and neighbor-joining (NJ) methods were also employed to approximate the distances and complete bootstrapping. As well as the general time reversible model assuming a discrete gamma distribution and invariant sites (GTR+I+G) for the combined aligned dataset, the Tamura-Nei model with gamma-distributed (TN93+G) for *rpb2* and the Kimura two parameter model (K2+G) for *tef1* were applied as the best evolutionary models for the phylogenetic analyses [25]. The topological support was then determined by 1000 bootstrap replicates. The sequences from the *Fusarium* spp. type strains, initially identified as closely related to the sequences here, were finally included by the preliminary BLAST searches (Table 1).

### 2.3. Pathogenicity Studies

To reproduce the dieback disease symptoms, the fungal isolates were tested for pathogenicity on the original host. A small, excised wound was accordingly made on the tip of each intact stem after being swabbed with ethanol 75% and washed with sterile water, then a mycelial agar disc (5 mm diameter) from each of the 7-day-old cultures of the fungal isolates was placed onto the surfaces of each stem tip and wrapped with Parafilm [8]. Afterward, the incubated plants were placed in a growth chamber at 25 °C and 75% relative humidity (RH) and maintained for 14 days. In contrast, the control plants received non-colonized agar plugs. Of note, the test was independently replicated thrice. All inoculated plants were visually assessed on a daily basis for up to two weeks. To fulfill Koch’s postulates, the same fungal isolates were re-isolated and their identity was confirmed by the *tef1* sequence data. To evaluate the disease severity, pear fruits (*Pyrus pyrifolia*) were also sterilized with 75% ethanol (3 min) and washed with sterile distilled water. Next, a mycelial agar disc (9 mm diameter) was placed on the fruit surface and covered with Parafilm to maintain high humidity [28]. All inoculated fruits were incubated under the same condition as mentioned above for one week. The fresh PDA agar plugs were further used as a negative control. The fruit rot diameter was finally measured by an electronic caliper seven days post inoculation (DPI). Each treatment included three replicates, and the experiment was independently repeated at least twice for both tests.

## 3. Results

### 3.1. Field Survey, Disease Symptoms, and Pathogen Isolations

In September 2021, symptoms of the dieback disease on *D. officinale* emerged in Yueqing, Zhejiang Province, China. According to the field observations, high temperature, high humidity, and poor ventilation would accelerate the incidence rate of this condition, which was at about 40–60% based on the number of plants with dieback disease symptoms recorded in 30 rows randomly picked. The symptoms appeared as chlorotic, blighted, and wilted leaves of the apical meristem with the shoot tip showing dark brown necrosis, dieback, and eventually shoot death (Figure 1). A total of 152 *Fusarium*-like isolates were also recovered from 70 infected plants, and 20 representative isolates were selected for further analysis (Table 2). Each isolate was recovered from different infected stems. Consistent with their morphological traits as well as molecular methods, the isolated fungi belonged to five genera, encompassing *F. concentricum*, *F. curvatum*, *F. fujikuroi*, *F. nirenbergiae*, and *F. stilboides*. Comparing the isolation frequency accordingly revealed that *F. concentricum* was the most abundant species, followed by *F. curvatum*, *F. fujikuroi*, and *F. nirenbergiae*, while *F. stilboides* was found the least (Table 2). Interestingly, two, and occasionally more than two, different *Fusarium* spp. were simultaneously isolated from some samples, and finally confirmed by the *tef1* and *rpb2* sequence analyses.

### 3.2. Morphological Identification

The phenotypic criteria of the representative isolates obtained from the symptomatic stem tips matched with the descriptions of *F. concentricum*, *F. fujikuroi*, *F. nirenbergiae*, *F. curvatum*, and *F. stilboides* morphology.

In this regard, *F. concentricum* showed yellow-white, abundant, densely lanose to velutinous aerial hyphae with concentric rings on the PDA. Also, the colonies produced mainly 3-5-septate, naviculate to fusiform, slender macroconidia (Sporodochial conidia) with beaked apical and foot-shaped basal cells. Microconidia (aerial conidia) were also obovoid to fusoid, predominantly with no septa, but occasionally with 1 septum, and borne on mono or poly-phialides found in the aerial mycelia. However, chlamydospores were not observed. On the CLA, orange sporodochia were found (Figure 2).

On the PDA, *F. curvatum* formed abundant floccose aerial mycelium with pale rosy white hue. Microconidia were also hyaline, ellipsoidal to falcate, 0-1-septate, and bore forming small false heads (i.e., short unbranded conidiophores) on the tips of the phialides. Besides, macroconidia were hyaline, 2-4-septate, banana-shaped, with blunt to papillate apical and blunt basal cells. Also, chlamydospores were not observed, and orange sporodochia formed on the carnation leave (Figure 2).

The *F. fujikuroi* colonies on the PDA consisted of floccose white aerial mycelia that became gray-violet or magenta with age, lacking chlamydospores. Notably, some swollen cells could develop in the hyphae and superficially appear chlamydospores or pseudochlamydospores. Aerial conidia were also oval-shaped with a flattened base and 0-1-septate on the CLA. The long, slender, usually 3–6-septate macroconidia further proliferated on the monophialides of the branched conidiophores in the sporodochia. Moreover, the pale orange sporodochia was sparsely produced on the CLA (Figure 2).

The *F. nirenbergiae* colonies were pale vinaceous to burly-wood color, with abundant flocculent aerial hyphae on the PDA. Sporodochial conidia also formed small false heads on the tips of the phialides, lucid, oval to falcate with 0-1-septa. As well, macroconidia were hyaline, generally 3-septate, and in the shape of crescents or sickles with an attenuated to semi-papillate, curved apical and foot-shaped basal cells. The globose to spherical, aseptate chlamydospores were further produced terminally or intercalary. The aerial mycelium also formed abundantly bright orange sporodochia on the CLA (Figure 2).

The aerial mycelia of *F. stilboides* strains were cottony, velvety, reddish orange to maroon on the PDA. The aerial conidia were also long, cylindrical, smooth-walled, 3-5-septate, straight to almost slightly flexuous in the center and sharpened at the apices with marked foot-shaped cells. Moreover, formation of orange sporodochia was observed on CLA. The microconidia were typically obovoid to elliptical and 0-1-septate, and chlamydospores were present (Figure 2).

### 3.3. Molecular Identification and Phylogenetic Analyses

The PCR amplified partial sequences of the genes *tef1*, *rpb2*, and *tub2*, yielded 651, 718, and 481 bp fragments, respectively. The BLASTN searches against all *Fusarium* sequences in the GenBank additionally showed that the 20 representative isolates included in this study shared 99–100% similarity with type-strains of five *Fusarium* spp., namely, *F. concentricum*, *F. fujikuroi*, *F. nirenbergiae*, *F. curvatum*, and *F. stilboides*, which supported previous efforts for the identification of these pathogens based on the macro and micro-morphological characteristics. To clarify the phylogenetic relations, the phylogenetic trees were built here from the single genes *tef1* and *rpb2*. These trees included several sequences from the new isolates, *Fusarium* type strains, plus a few non-type strains. The given trees further supported the identification of the *D. officinale* isolates (Appendix A). The partial *tub2* sequences also displayed close similarity with the *Fusarium* species but provided an insufficient resolution to identify them. For further molecular verification, multilocus phylogenetic analysis (MLSA) was further performed based on 1850 nucleotide positions among 91 in-group taxa, including clades corresponding to FOSC, FFSC, FLSC, and FIESC. The MLSA tree accordingly indicated that the D. officinale isolates in the present study clustered unambiguously with *F. concentricum*, *F. fujikuroi*, *F. nirenbergiae*, *F. curvatum*, and *F. stilboides* type strains with the bootstrap values of 99%, 99%, 96%, 95%, and 99%, respectively (Figure 3). The topologies of the trees obtained from each individual gene also resembled each other and were, above all, similar to the MLSA tree (Appendix A). Nonetheless, the concatenated dataset, in addition to all the individual phylogenies, clearly determined the phylogenetic relationship and taxonomy of Tiepishihu isolates in the Fusariod taxa. Moreover, the *F. equiseti* strain DFS, as a pathogen of the dieback disease of *D. officinale* in China, fell strongly with the *F. equiseti* clade, which belonged to the FIESC (Appendix A).

### 3.4. Pathogenicity Assays

Two weeks after inoculation, the pathogenicity test results revealed that the isolates from five *Fusarium* spp. had the typical black brown necrosis and the dieback symptoms on the tip of *D. officinale* (tie-pie variety) stems, which were congruent with the field observations, while no symptoms developed on the control plants inoculated with the agar media. All these fungal species were also re-isolated from the inoculated plants, and identified using the *tef1* locus, thereby fulfilling Koch’s postulates (Figure 4A). Notably, the dieback disease symptoms incited by each pathogen were indistinguishable in the field. As such, all *Fusarium* spp. isolates were pathogenic on Tiepishihu and caused the dieback disease in these inoculation studies. Furthermore, *F. concentricum* and *F. fujikuroi*, among the assayed isolates, indicated higher virulence on pears than the others, followed by *F. nirenbergiae* and *F. curvatum*, which had relatively similar disease severity, whereas *F. stilboides* showed the lowest virulence in this respect (Figure 4B; Table 3).

## 4. Discussion

The dieback disease has already plagued the *D. officinale* industry with a high incidence rate in Zhejiang Province, China. On the basis of the MLSA supported by morphological observations in this study, five distinct taxa, viz., *F. concentricum*, *F. fujikuroi*, *F. nirenbergiae*, *F. curvatum*, and *F. stilboides*, causing the dieback disease on Tiepishihu, were diagnosed. The study findings also suggested that the *Fusarium* species associated with this condition on Tiepishihu were more diverse than the ones previously recorded [8]. The Koch’s postulates correspondingly showed that the *Fusarium* spp. Isolates were infective in nature, with slight variations in virulence.

The newly inflicted *Fusarium* spp. On *D. officinale* have been also found as plant pathogens and innocuous saprophytes, e.g., *F. concentricum* causing fruit blotch on *Hibiscus sabdariffa*, stem rot on *Paris polyphylla*, fruit rot on pepper and banana, leaf spot on mango, wilt on *Podocarpus macrophyllus*, and ear rot on maize, originally named by Nirenberg and O’Donnell [24,29,30,31,32]. *F. curvatum* has been also described as a new taxon by Lombard et al. [16], originally named *F. oxysporum* (*matthiolae* and *meniscoideum formae speciales*), which was regarded as a FOSC member. However, little is known about this species, and it has been reported as a pathogen on yam [13,16,24,33]. In addition, *F. fujikuroi* is a well-studied taxon with sexual stage, *Gibberella fujikuroi* (Sawada), which is known as a pathogen on various host plants including cotton, *Echinochloa* sp., grapes, maize, *Macleaya cordata*, *Reineckia carnea*, rice, soybean, strawberries, sugarcane, wheat, and *Zanthoxylum armatum* [24,32,34]. Moreover, *F. nirenbergiae* was recently resolved from the FOSC and included several *formae speciales* of *F. oxysporum* (viz., *dianthi*, *chrysanthemi*, *bouvardiae*, *adices-lycopersici*, *cubense*, *lycopersici*, and *passiflorae*) and clinically relevant strains [13,16,24]. This pathogen has been further associated with multiple diseases such as saffron corm rot, wilt on *Acer negundo*, *Dipladenia* sp., and passion fruit [16,35,36,37]. *Fusarium stilboides* is an FLSC member that has been recorded as a pathogen on bamboo, *Capsicum annuum*, carnation, coffee, and passion fruit [13,38,39,40,41].

Although macro- and micro-morphological observations alone may be insufficient, several critical characteristics can provide useful information for discriminating the *Fusarium* species [13,14,15,16]. Hence, the detailed and close morphological examinations form an important part in the classification of this genus. In this line, *F. curvatum* produces high curvature macroconidia and aerial polyphialidic conidiogenous cells, differentiating it from *F. nirenbergiae* and other species [13,14,16]. Moreover, chlamydospores are readily present in *F. nirenbergiae*, while this feature is not known for *F. concentricum*, *F. fujikuroi*, and *F. curvatum* [16,17,18,19,20,21,22,23,24]. However, *F. stilboides* can produce chlamydospores even though this trait is not taxonomically useful [24]. Additionally, *F. nirenbergiae* cannot form polyphialides, and resembles *F. oxysporum* in this respect [16,17,18,19,20,21,22,23,24]. Morphologically, both *F. fujikuroi* and *F. concentricum* proliferate microconidia in false heads or chains, whereas the conidiophores of the aerial mycelium are only sparsely branched in *F. concentricum* with only a few polyphialides, and conidia in *F. fujikuroi* conidia are from polyphialides, and less often from monophialides [24,30,32].

The validity of morphological identification in this study was thus confirmed by the phylogenetic analyses derived from the molecular results. The *tef1*, *rpb2*, and *tub2* genetic barcodes were thus selected for this purpose because they consisted of phylogenetically informative sequences for the differentiation and classification of *Fusarium* species [13,16,19,20].

The *tef1* phylogeny accordingly demonstrated better resolution at the species level in comparison with *rpb2* and *tub2*. For the concatenated gene analyses, the topologies of the trees inferred for individual genes were also evaluated visually to establish that the overall tree topology of the single locus datasets were similar to each other and to that of the tree acquired from the combined dataset alignment. In the MLSA tree, the *Fusarium* isolates from *D. officinale* in the present study were phylogenetically different from each other and were situated within the FOSC, FFSC, and FLSC clades with *F. concentricum* CBS 450.97, *F. curvatum* CBS 238.94, *F. fujikuroi* CBS 221.76, *F. nirenbergiae* CBS 840.88, and *F. stilboides* CBS 746.79 type strains, while none of them fell in the FBSC, FRSC, and FsaSC clades. Even if *F. nirenbergiae* and *F. curvatum* were placed in the same clade, they formed two distinct well-supported subclades, which correlated with clade VIII resolved by Lombard et al. [16]. Furthermore, the tree topology of the concatenated dataset in this study was closely similar to the trees developed by Lombard et al. [16], Crous et al. [13], and Yilmaz et al. [19].

Interestingly, it has been concluded that the dieback is a disease complex, induced by one or more *Fusarium* spp. (viz., *F. concentricum*, *F. curvatum*, *F. fujikuroi*, *F. nirenbergiae*, and *F. stilboides*), as observed in manifold crops [33,42,43,44]. For instance, eight species including *F. asiaticum*, *F. equiseti*, *F. fujikuroi*, *F. graminearum*, *F. meridionale*, *F. oxysporum*, *F. proliferatum*, and *F. verticillioides*, have been evidenced to incite corn sheath rot in Sichuan Province, China [44]. Similarly, ten *Fusarium* species, viz., *F. asiaticum*, *F. commune*, *F. cugenangense*, *F. curvatum*, *F. fujikuroi*, *F. gossypinum*, *F. nirenbergiae*, *F. odoratissimum*, *F. solani* and *F. verticillioides*, have been detected to cause wilt on yam [33].

Despite much effort, there was no success in defining the pathogen-specific diagnostic criteria for these five taxa, which induced the dieback disease on *D. officinale*. Further surveys are thus required to establish the species-specific symptoms of this condition. Additionally, the symptoms on some stem tips may slightly differ in respect of intensity or color, suggesting that such symptoms may have been due to secondary infections by saprophytic microbes or affected by environmental conditions, such as high humidity or unfavorable ventilation, as documented in previous research [21,22,35,45,46,47,48,49,50].

## 5. Conclusions

In sum, the study data here confirmed that the losses in *D. officinale* yields were caused by *F. concentricum*, *F. curvatum*, *F. fujikuroi*, *F. nirenbergiae*, and *F. stilboides*. Notably, the incidence rate of these local outbreaks could be triggered by environmental factors, and it is expected to increase in the future because of both climate change and susceptible cultivars. Regarding the significance of this study, it provided information on the biodiversity and epidemiology of *Fusarium* spp. associated with the dieback disease, which can contribute to the development of breeding programs and disease management strategies.

## Figures and Tables

**Figure 1 jof-08-00919-f001:**
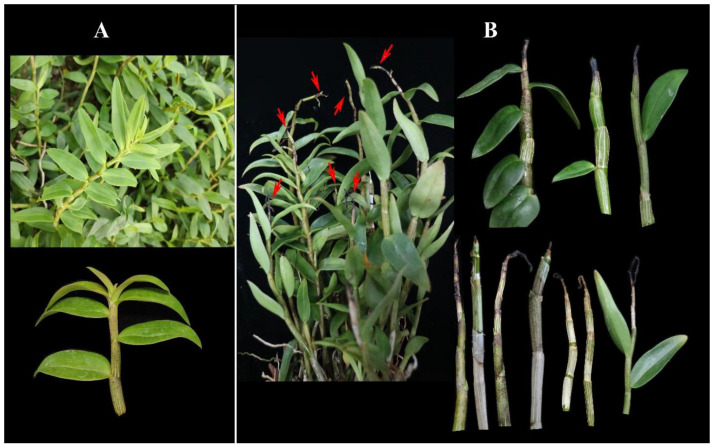
Natural symptoms on *Dendrobium officinale* tissue associated with *Fusarium* spp. (**A**) Healthy plant; (**B**) typical dieback on the stem.

**Figure 2 jof-08-00919-f002:**
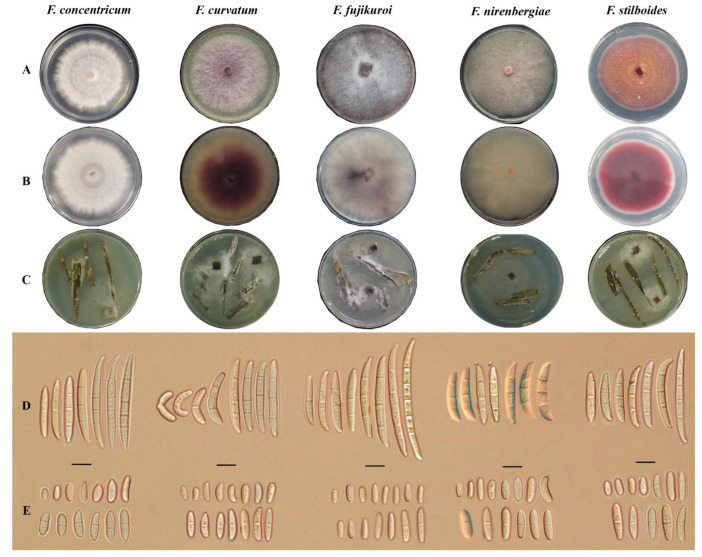
Morphological characteristics of *Fusarium* species isolated from infected *Dendrobium officinale* stems. (**A**) Front side on PDA; (**B**) Reverse side on PDA; (**C**) Sporodochia on Carnation Leaf Agar; (**D**) Macroconidia; (**E**) Microconidia. Scale bars: D–E = 10 µm.

**Figure 3 jof-08-00919-f003:**
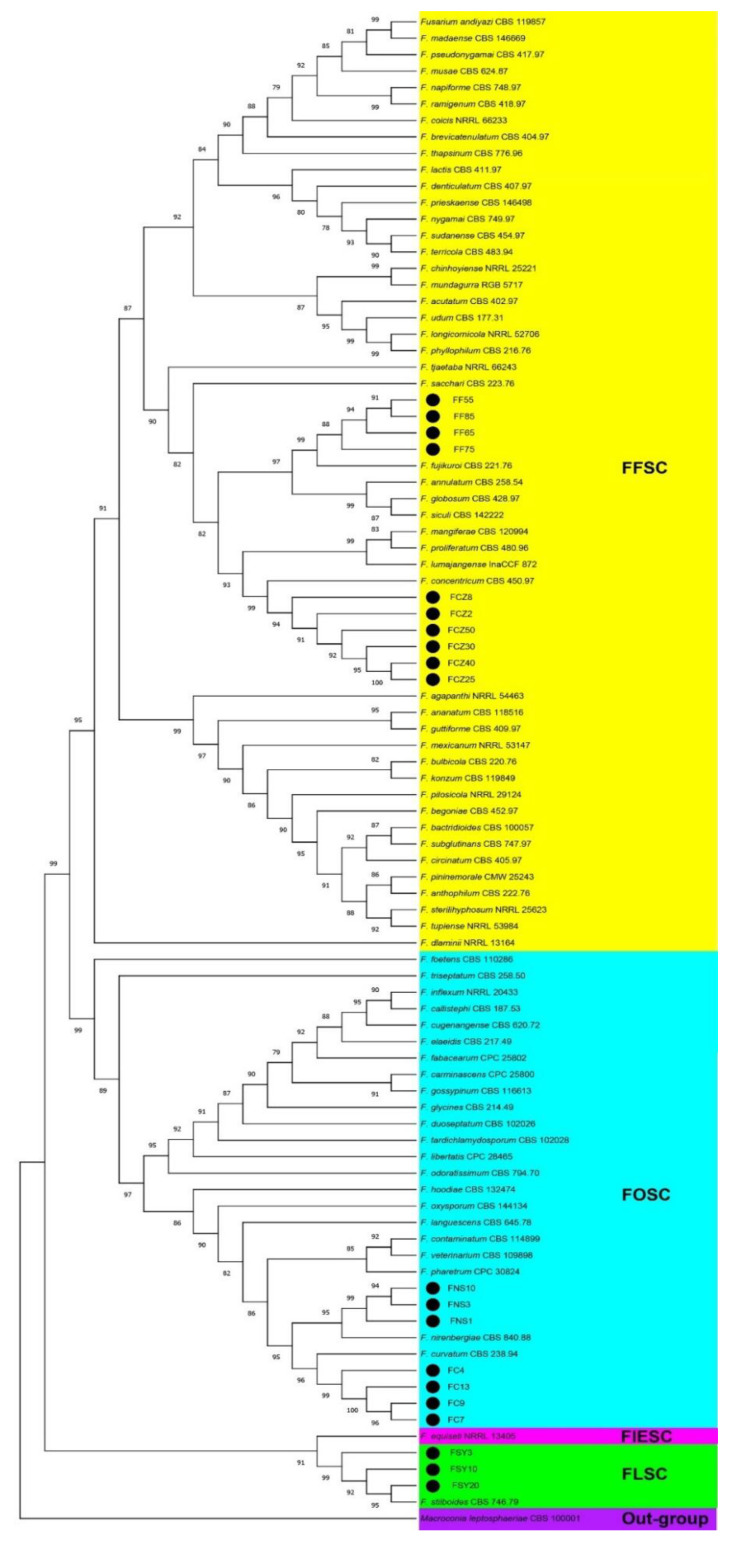
Multilocus phylogenetic tree resulting from maximum likelihood analysis of concatenated *rpb2*, *tef1*, and *tub2* sequences. The tree shows the phylogenetic relationships of *Fusarium* spp. causing dieback disease in *Dendrobium officinale*. Isolates recovered from Tiepishihu in this study are indicated by a black circle (●). Clades including isolates obtained from *D. officinale* are shaded in color. The tree is rooted to *Macroconia leptosphaeriae* CBS 100001. Bootstrap values are shown above the branches. Subdivision of the *Fusarium* clade represents the recognized species complexes.

**Figure 4 jof-08-00919-f004:**
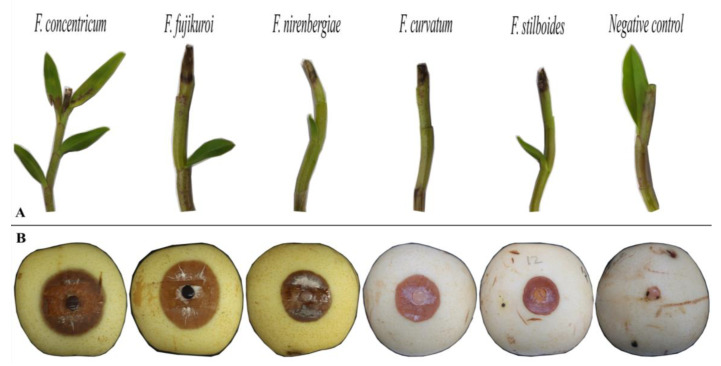
Artificial symptoms on tip of *Dendrobium officinale* stems (**A**), and *Pyrus pyrifolia* fruits (**B**) after inoculation with *Fusarium* species.

**Table 1 jof-08-00919-t001:** GenBank accession numbers of *Fusarium* strains used in the phylogenetic analyses.

Species	Culture Collection No./Isolate	GenBank Accession		
		*rpb2*	*tef1*	*tub1*
*Fusarium carminascens*	CPC 144738 T	MH484937	MH485028	MH485119
*F. contaminatum*	CBS 111552 T	MH484901	MH484992	MH485083
*F. pharetrum*	CBS 144751 T	MH484952	MH485043	MH485134
*F. veterinarium*	CBS 109898 T	MH484899	MH484990	MH485081
*F. cugenangense*	CBS 620.72	MH484879	MH484970	MH485061
*F. curvatum*	CBS 238.94 T	MH484893	MH484984	MH485075
	**FC4**	**ON137565**	**ON137544**	**ON137586**
	**FC6**	**ON137566**	**ON137545**	**ON137587**
	**FC7**	**ON137567**	**ON137546**	**ON137588**
	**FC9**	**ON137568**	**ON137547**	**ON137589**
	**FC13**	**ON137568**	**ON137548**	**ON137590**
*F. fabacearum*	CPC 25802 T	MH484939	MH485030	MH485121
*F. glycines*	CBS 144746 T	MH484942	MH485033	MH485124
*F. gossypinum*	CBS 116613 T	MH484909	MH485000	MH485091
*F. languescens*	CBS 645.78 T	MH484880	MH484971	MH485062
*F. libertatis*	CPC 28465 T	MH484944	MH485035	MH485126
*F. nirenbergiae*	CBS 840.88 T	MH484887	MH484978	MH485069
	**FNS1**	**ON137574**	**ON137553**	**ON137595**
	**FNS3**	**ON137575**	**ON137553**	**ON137596**
	**FNS10**	**ON137576**	**ON137553**	**ON137597**
*F. oxysporum*	CBS 144134 ET	MH484953	MH485044	MH485135
*F. hoodiae*	CBS 132474 T	MH484929	MH485020	MH485111
*F. duoseptatum*	CBS 102026 T	MH484896	MH484987	MH485078
*F. callistephi*	CBS 187.53 T	MH484875	MH484966	MH485057
*F. triseptatum*	CBS 258.50 T	MH484910	MH485001	MH485055
*F. languescens*	CBS 645.78 T	MH484880	MH484971	MH485062
*F. elaeidis*	CBS 217.49	MH484870	MH484961	MH485052
*F.* *acutatum*	CBS 402.97 T	MW402768	MW402125	MW402323
*F. agapanthi*	NRRL 54463 T	KU900625	KU900630	KU900635
*F. ananatum*	CBS 118516 T	LT996137	LT996091	MN534089
*F. andiyazi*	CBS 119857 T	LT996138	MN193854	LT996113
*F. annulatum*	CBS 258.54 T	MT010983	MT010994	MT011041
*F. anthophilum*	CBS 222.76 ET	MW402811	MW402114	MW402312
*F.* *bactridioides*	CBS 100057 T	MN534235	MN533993	MN534112
*F. begoniae*	CBS 452.97 T	MN534243	MN533994	MN534101
*F. brevicatenulatum*	CBS 404.97 T	MN534295	MN533995	MN534063
*F. bulbicola*	CBS 220.76 T	MW402767	KF466415	KF466437
*F. chinhoyiense*	NRRL 25221 T	MN534262	MN534050	MN534082
*F. subglutinans*	CBS 747.97 NT	MW402773	MW402150	MW402351
*F. circinatum*	CBS 405.97 T	MN534252	MN533997	MN534097
*F. coicis*	NRRL 66233 T	KP083274	KP083251	LT996115
*F. concentricum*	CBS 450.97 T	JF741086	AF160282	MW402334
	**FCZ2**	**ON137559**	**ON107278**	**ON137580**
	**FCZ8**	**ON137560**	**ON107279**	**ON137581**
	**FCZ25**	**ON137561**	**ON107280**	**ON137582**
	**FCZ30**	**ON137562**	**ON107281**	**ON137583**
	**FCZ40**	**ON137563**	**ON107282**	**ON137584**
	**FCZ50**	**ON137564**	**ON107283**	**ON137585**
*F. globosum*	CBS 428.97 T	KF466406	KF466417	MN534124
*F. guttiforme*	CBS 409.97 T	MT010967	MT010999	MT011048
*F. konzum*	CBS 119849 T	MW402733	LT996098	MN534095
*F. lactis*	CBS 411.97 ET	MN534275	MN193862	MN534077
*F. longicornicola*	NRRL 52706 T	JF741114	JF740788	MW402360
*F. denticulatum*	CBS 407.97 T	MN534274	MN534000	MN534068
*F. dlaminii*	CBS 119860 T	KU171701	MW401995	MW402195
*F. fujikuroi*	CBS 221.76 T	KU604255	MN534010	MN534130
	**FF55**	**ON137570**	**ON137549**	**ON137591**
	**FF65**	**ON137571**	**ON137550**	**ON137592**
	**FF75**	**ON137572**	**ON137551**	**ON137593**
	**FF85**	**ON137573**	**ON137552**	**ON137594**
*F.* *madaense*	CBS 146669 T	MW402764	MW402098	MW402297
*F.* *mangiferae*	CBS 120994 T	MN534271	MN534017	MN534128
*F.* *mexicanum*	NRRL 53147 T	MN724973	GU737282	GU737494
*F.* *mundagurra*	RGB5717 T	KP083276	KP083256	MN534146
*F.* *musae*	CBS 624.87 T	MW402772	FN552086	FN545368
*F. napiforme*	CBS 748.97 T	MN534291	MN193863	MN534085
*F. nygamai*	CBS 749.97 T	EF470114	MW402151	MW402352
*F. phyllophilum*	CBS 216.76 T	KF466410	MN193864	KF466443
*F. pilosicola*	NRRL 29124 T	MN534248	MN534055	MN534099
*F. proliferatum*	CBS 480.96 ET	MN534272	MN534059	MN534129
*F. pseudonygamai*	CBS 417.97 T	MN534285	AF160263	MN534066
*F. ramigenum*	CBS 418.97 T	KF466412	KF466423	MN534145
*F. sacchari*	CBS 223.76 ET	JX171580	MW402115	MW402313
*F.* *siculi*	CBS 142222 T	LT746327	LT746214	LT746346
*F.* *succisae*	CBS 219.76 ET	MW402766	AF160291	U34419
*F.* *sudanense*	CBS 454.97 T	MN534278	MN534037	MN534073
*F.* *terricola*	CBS 483.94 T	LT996156	MN534042	MN534076
*F* *.thapsinum*	CBS 776.96 T	MN534289	MN534044	MN534080
*F.* *tjaetaba*	NRRL 66243 T	KP083275	KP083263	GU737296
*F. tupiense*	NRRL 53984	LR792619	GU737404	GU737296
*F. prieskaense*	CBS 146498 T	MW834006	MW834274	MW834302
*F. foetens*	CBS 110286 T	MW928825	MT011001	MT011049
*F. hostae*	NRRL 29889 T	MT409446	MT409456	AY329042
*F. udum*	NRRL 25199 ET	KY498875	KY498862	KY498892
*F. stilboides*	CBS 746.79 T	MW928832	MW928843	-
	**FSY3**	**ON137577**	**ON137556**	**ON137598**
	**FSY10**	**ON137578**	**ON137557**	**ON137599**
	**FSY20**	**ON137579**	**ON137558**	**ON137600**
*F. buharicum*	NRRL 25488	KX302928	KX302912	-
*F. equiseti*	NRRL 26419	-	GQ505599	-
	NRRL 20697	-	GQ505594	-
	NRRL 13405 T	GQ915491	GQ915507	GQ915441
	DFS	**-**	MN823983	-
*F.* *peruvianum*	CBS 511.75	**-**	MN120767	-
*F. sarcochroum*	CBS 745.79	JX171586	MW834278	-
*F. inflexum*	NRRL 20433	JX171583	AF008479	U34435
*F. sublunatum*	NRRL 20897	KX302935	KX302919	-
*F. convolutans*	CBS 144207 T	LT996141	LT996094	-
*F. sarcochroum*	CBS 745.79	-	MW834278	-
*Macroconia leptosphaeriae*	CBS 100001	HQ728164	KM231959	KM232097

CBS: Westerdijk Fungal Biodiversity Institute (WIFB), Utrecht, The Netherlands. NRRL (Northern Regional Research Laboratory): Agricultural Research Service Culture Collection Database, Peoria, USA. CMW: The working collection of FABI (Forestry and Agricultural Biotechnology Institute), University of Pretoria, South Africa. BBA: Julius Kühn-Institute, Institute for Epidemiology and Pathogen Diagnostics, Berlin and Braunschweig, Germany. CPC: Collection of P.W. Crous. T: Ex-type specimen. NT: Neotype specimen. ET: Ex-epitype specimen. Accession numbers in **bold** belong to newly determined *Dendrobium officinale* isolates of *Fusarium* spp.

**Table 2 jof-08-00919-t002:** Sampling details, number of isolates collected, and frequency of fungal species identified in the present study.

Geographic Origin	Species	No. Isolate	Isolation Frequency (%)
Zhejiang Province (Yueqing City)	*F. concentricum*	53	34.6
	*F. curvatum*	34	22.3
	*F*. *fujikuroi*	28	18.4
	*F. nirenbergiae*	21	13.8
	*F. stilboides*	16	10.5
	Total	152	-

**Table 3 jof-08-00919-t003:** Virulence testing of *Fusarium* species on *Pyrus pyrifolia*.

Species	Mean Lesion Size (mm) ± SD
*F. concentricum*	45.30 ^a^ ± 3.19
*F*. *fujikuroi*	41.83 ^b^ ± 1.84
*F. nirenbergiae*	39.85 ^c^ ± 1.64
*F. curvatum*	34.8 ^d^ ± 1.66
*F. stilboides*	28.92 ^e^ ± 2.00

Note: The severity of the isolates was assessed by measuring the length of the fruit rot on *P. pyrifolia* at 7 DPI in two perpendicular directions. Data are mean ± SE. The mean values followed by different letters represent significantly different values at *p* < 0.05 among species using the least significant difference test (LSD).

## Data Availability

Sequences have been deposited in GenBank (Table 1). The data presented in this study are openly available in NCBI. Publicly available datasets were analyzed in this study. These data can be found here: https://www.ncbi.nlm.nih.gov/ (accessed on 2 April 2022).

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
