# Peer review of "Fusarium spp. Associated with Dendrobium officinale Dieback Disease in China"

_jof, 2022, doi:10.3390/jof8090919_

Round 1

Reviewer 1 Report

Which path is the death of plants: necrosis or PCD? It must be confirmed by tests.

The stability of the fungus must be confirmed by different methods of analysis. For example, the determination of plant cell death by histochemical methods.

 What are the test methods for early diagnosis of plants for fusarium?

What are the proposed ways to increase the resistance of plants to fusarium?

 The authors stated about the optimization of strategies to combat the disease, what is it?

 It is necessary to check the list of references for compliance with the requirements of the journal.

Author Response

Dear Referee,

We sincerely express our deepest gratitude for your comments and gave us this chance to publish our paper in the Journal of Fungi. Furthermore, we carefully went through the whole manuscript to correct errors of grammar and improve the accuracy and clarity of the English, and added further literature citations. We hope the revised manuscript is now suitable for publication.

Regards

Ali

Comments and Suggestions for Authors

Reviewer: “Which path is the death of plants: necrosis or PCD? It must be confirmed by tests”.

Our response: Thank you for your good question. The tips of the stems were black brown necrosis and dieback and withered. inoculated Plants developed the same symptoms which we observed in the field.

Reviewer: “The stability of the fungus must be confirmed by different methods of analysis. For example, the determination of plant cell death by histochemical methods”.

Our response: Dear Professor, I cannot understand this comment. To confirm the ability of fungi whether pathogenic? Or not? we did “Koch’s postulates.” It is well-established approaches. We also did based on other published papers.

  1. The suspected causal agent must be present in every diseased organism (e.g., a plant) examined. 2. The suspected causal agent must be isolated from the diseased host organism (plant) and grown in pure culture. 3. When a pure culture of the suspected causal agent is inoculated into a healthy susceptible host (plant), the host must reproduce the specific disease. 4. The same causal agent must be recovered again from the experimentally inoculated and infected host, i.e., the recovered agent must have the same characteristics as the organism in step 2.

 Reviewer: “What are the test methods for early diagnosis of plants for fusarium?”

Our response: Currently, plant pathogen detection is heavily focused on a wide variety of molecular assays, including nucleic acid-based technologies such as PCR, loop-mediated isothermal amplification or DNA microarrays; and immunological approaches like enzyme-linked immunosorbent assays (ELISA). Nucleic acid-based diagnostics are the standard means for diagnosis of infected plant material. However, these methods are expensive and time-consuming, but they are accurate. Immunoassay technology offers simplicity and portability for on-site detection but is limited by detection sensitivity and specificity for certain applications. Alternatively, field-portable sensors have seen rapid development in the past few years and hold great promise. Non-invasive methods can be effectively used as reliable and accurate spectral instruments. Raman spectroscopy (RS) is label-free, non-destructive analytical method that can be used to confirm the diagnosis of biotic stresses in plants. A smartphone-integrated plant Volatile organic compound (VOC) profiling platform using a paper-based colourimetric sensor array that incorporates functionalized gold nanomaterials and chemo-responsive organic dyes for accurate and early detection. Some other VOC detection methods include Gas chromatography-mass spectrometry Selected ion flow tube–mas spectrometry, Proton transfer reaction mass spectrometry etc. Unfortunately, early detection of Fusarium diseases for most plants is very immature and still needs more work. Furthermore, non-invasive detection technologies and disease forecasting have not been established for Tiepishihu, further studies need to focus on these approaches for early detection.

Reviewer: “What are the proposed ways to increase the resistance of plants to fusarium?”

Our response: As you know, some diseases are complex diseases and are caused by several species. In this type of pathosystem, some species have higher prevalency and virulence ability than others (It is not mean that species with lower frequency have less importance than others). Moreover, many published papers mentioned that some species with low frequency (ratio) showed high virulence in comparison with some species with high prevalence. Here we should not ignore the role of environmental conditions. Therefore, it is common and predictable that after some time, we observe resistant cultivars lose their resistance to some diseases. One of the important reasons for this phenomenon is that during the process of producing resistant cultivars, we do not accurately identify all population's pathogens in a pathosystem. by applying a polyphasic approach we detect all cryptic species within different genus&species, leading to establishing a successful breeding program. Finally, Evaluate each isolated-pathogen reaction against various cultivars by visual observation and QTL Analysis for identifying resistance genes and introducing resistant cultivars. It is worth to mention that Integrated disease management strategies include biological control agents, biofertilizer, chemical fungicides, cultural practices, forecasting models, and sanitation methods are essential to minimize the incidence of plant pathogens.

Reviewer: “The authors stated about the optimization of strategies to combat the disease, what is it?”

Our response: thank you very much to remind us. We have remove it.

 Comments and Suggestions for Authors

 Reviewer: “ However, the authors must improve the English and the scientific writing of their manuscript to bring it to a very mature levels before get it accepted for publication”.

Our response: Dear Professor, Thank you very much. I have sent it to a good English company.

Reviewer: “Correct the isolated fungi belonged to 5 genera (5 species) wherever written.”

Our response: we have changed “five genera” to 5 genera”.

Reviewer: “Avoid mentioning too much about materials and methods in the introduction and focus the introduction on the study background”.

Our response: I am grateful to you. we have removed “materials and methods” from the last paragraph of the introduction.

Reviewer: “Support your discussion with some more references.”   

Our response: we have added more new references.

Reviewer 2 Report

  The manuscript 'Fusarium spp. Associated With Dendrobium officinale Dieback Disease in China' brings good new values to our knowledge. However, the authors must improve the English and the scientific writing of their manuscript to bring it to a very mature levels before get it accepted for publication. Correct the isolated fungi belonged to 5 genera (5 species) wherever written. Avoid mentioning too much about materials and methods in the introduction and focus the introduction on the study background. Support your discussion with some more references.   

Author Response

Dear Referee,

We sincerely express our deepest gratitude to your comments and gave us this chance to publish our paper in the Journal of Fungi. Furthermore, we carefully went through the whole manuscript to correct errors of grammar and improve accuracy and clarity of the English, and added further literature citations. We hope the revised manuscript is now suitable for publication.

Regards

Ali

 Reviewer: “ However, the authors must improve the English and the scientific writing of their manuscript to bring it to a very mature levels before get it accepted for publication”.

Our response: Dear Professor, Thank you very much. I have sent it to a good English company.

Reviewer: “Correct the isolated fungi belonged to 5 genera (5 species) wherever written.”

Our response: we have changed “five genera” to :5 genera”.

Reviewer: “Avoid mentioning too much about materials and methods in the introduction and focus the introduction on the study background”.

Our response: I am grateful to you. we have removed “materials and methods” from the last paragraph of the introduction.

Reviewer: “Support your discussion with some more references.”   

Our response: we have added more new references.

Reviewer 3 Report

Dear Authors

The manuscript with this title; 

Fusarium spp. Associated With Dendrobium officinale Dieback Disease in
China

is a benefit research manuscript for readers, but before publishing needs to improve by some corrections.

In introduction part add some new research in recent years.

before final paragraph of introduction, add some researches about control of plant pathogens such as below reference:

Encapsulation of Plant Biocontrol Bacteria with Alginate as a Main Polymer Material

Roohallah Saberi Riseh , Yury A. Skorik , Vijay Kumar Thakur, Mojde Moradi Pour , Elahe Tamanadar  and Shahnaz Shahidi Noghabi

The discussion section needs to be improved.

Please write the importance of this research clearly.

Add some new references in discussion part.

Totally, I will suggest this manuscript for publication after minor revision.

Author Response

Dear Referee,

We sincerely express our deepest gratitude for your comments and gave us this chance to publish our paper in the Journal of Fungi. Furthermore, we carefully went through the whole manuscript to correct errors of grammar and improve accuracy and clarity of the English and added further literature citations. We hope the revised manuscript is now suitable for publication.

Regards

Ali

Reviewer: “before final paragraph of introduction, add some researches about control of plant pathogens such as below reference”.

Our response: we have added more references.

Reviewer: “Please write the importance of this research clearly”.

Our response: We have mentioned the importance of this research in the Conclusion part.

Reviewer: “Add some new references in discussion part”.

Our response: we have added more new references.

Reviewer 4 Report

Dear colleagues,

This review is concerning a research work entitled “Fusarium spp. Associated With Dendrobium officinale Dieback Disease in China

The major points are:

1-    Conclusion should be explain more than this in Abstract even in the end of discussion

3-    What is the variety of  plant used here and it source

4-    English language should be revise well

5-    Put all Latin names in italics throughout the text (it is wrong in quite many places, check them one by one); put also the names of the author(s) of all taxa cited the first time they appear in the text

6-    Keyword should be different than in title

7- Make sure that all scientific names in the References list are italics.

8- Please add the DOI for ALL the References

9- All tables must be self-explanatory.

10- you should not used references  in results  [13,16,19,22,28].

11- why you divided the table 1 in two part even it was in one page 

12- where is statistical analysis here I did not read anything about it 

13- the same Fig and table in the text was added as supplementary material 

Author Response

Dear Referee,

We sincerely express our deepest gratitude for your comments and gave us this chance to publish our paper in the Journal of Fungi. Furthermore, we carefully went through the whole manuscript to correct errors of grammar and improve the accuracy and clarity of the English, and added further literature citations. We hope the revised manuscript is now suitable for publication.

Regards

Ali

Reviewer: “Conclusion should be explain more than this in Abstract even in the end of discussion”

Our response: we have changed the Conclusion.

Reviewer: “What is the variety of  plant used here and it source”

Our response: the plant is produced by tissue culture and microprpagation method. The variety which cultured in our investigated area was “tie-pie”.

Reviewer: “ English language should be revise well”

Our response: Dear Professor, Thank you very much. I have sent it to a good English company..

Reviewer: “Put all Latin names in italics throughout the text “

Our response: we have made all Latin names in italics throughout the text

Reviewer: “put also the names of the author(s) of all taxa cited the first time they appear in the text”

Our response: we have added for “Dendrobium officinale” and “Fusarium”.

Reviewer: “Keyword should be different than in title”

Our response: We have changed the Keyword 

Reviewer: “Make sure that all scientific names in the References list are italics”.

Our response: we have made all Latin names in italics in References

Reviewer: “ Please add the DOI for ALL the References”

Our response: we have put DOI for all references

Reviewer: “you should not used references in results [13,16,19,22,28]”.

Our response: we have removed it.

Reviewer: “why you divided the table 1 in two part even it was in one page” 

Our response: well, because of data was too much. I could not put them in one page. Now the journal put them in one table.

Reviewer: “where is statistical analysis here I did not read anything about it “

Our response: we have changed it and added new statistical analysis data

Reviewer: “the same Fig and table in the text was added as supplementary material” 

Our response: we have only one Fig which should be as supplementary material, the other table and Figs belong to the main manuscript.

Round 2

Reviewer 2 Report

The authors did revised and improve their manuscript [ Fusarium spp. Associated With Dendrobium officinale Dieback Disease in China) and therefore I recommend publishing it.

Author Response

Dear Senior Editors and Referees,

I express my deepest gratitude for your support. The manuscript was revised with another native English Speaker in addition to the English Company.

I hope our paper now deserves to be published in the Journal.

Best Wishes

Ali